# Serum Paraoxonase-1 Activity in Prostate Cancer Patients Treated with Brachytherapy as a Measure of Irradiation Efficacy

**DOI:** 10.3390/antiox12020212

**Published:** 2023-01-17

**Authors:** Dorota Olszewska-Słonina, Miłosz Jasiński

**Affiliations:** 1Department of Pathobiochemistry and Clinical Chemistry, Ludwik Rydygier Collegium Medicum in Bydgoszcz, Nicolaus Copernicus University, 87-100 Toruń, Poland; 2Department of Urology and Urological Oncology, Collegium Medicum, University of Zielona Góra, 65-417 Zielona Góra, Poland

**Keywords:** paraoxonase-1, prostate cancer, radiotherapy

## Abstract

We investigated changes in the activity of antioxidant paraoxonase-1 (PON1) in patients with prostate cancer (PCa) undergoing radiotherapy (RT), as well as the relationship of the PON1 activity with the degree of PCa advancement. We included 84 men with PCa. Blood samples were obtained before irradiation and after the completion of RT. The control group was composed of 60 healthy men. There was no significant difference in the PON1 activity between the control group and patients pre-radiotherapy. Irradiation was associated with a significant decrease in the PON1 activity; thus, it could be a measure of the efficacy of RT. No significant correlations between the PON1 activity and Gleason score, prostate volume, BMI (body mass index), or adipose tissue thickness were found. However, there was a positive correlation between the PON1 activity and the PSA concentration in the group of PCa patients.

## 1. Introduction

Currently, there are two basic methods for the radical treatment of PCa confined to the organ: surgery (radical prostatectomy) and RT. RT can be carried out using two methods: external beam radiation therapy (EBRT) or intratissue irradiation–brachytherapy (BT); it is also possible to combine both methods (BT + EBRT). Brachytherapy uses radioactive sources implanted in the prostate parenchyma in the immediate vicinity of the tumor: permanent low-dose rate (LDR) brachytherapy using isotopes of iodine, palladium, and cesium (I-125, Pd-103, and Cs-131) or temporary high-dose rate (HDR) brachytherapy using the iridium isotope Ir-192 [1,2].

The basic criteria taken into account when qualifying patients for PCa brachytherapy are the PSA (prostate-specific antigen) concentration, tumor differentiation according to the Gleason classification, and TNM clinical stage [3,4,5]. Patients with baseline PSA levels less than 10 ng/mL, Gleason scores of 5–6, and T1-T2a disease are good candidates for BT as a stand-alone method [6]. Other patients usually require complementary hormonal treatment or teleradiotherapy, or they are not eligible for BT [7]. Local and distant metastases (N and M features) are the basic contraindication to the use of BT. The limitations may also include a large volume of the prostate gland (over 60 mL), symptoms of difficult urination (due to the increased risk of complications from the lower urinary tract associated with a radiation reaction in the urethra and bladder), and a history of transurethral resection of the prostate [2]. Three to six months of neoadjuvant hormone therapy may reduce the volume of the prostate below 60 mL and enable the use of BT [7]. Relative contraindications include the high location of the gland under the pubic symphysis, inflammation within it, mild hypertrophy of the middle lobe, and inflammatory diseases of the intestines and rectum [2,4].

HDR (high-dose rate) brachytherapy is a method of PCa treatment with an effectiveness that has been confirmed in many studies, as well as a five-year biochemical recurrence-free survival rate of 90, 85, and 65%, respectively, for low-, medium-, and high-risk groups [5,7].

Reactive oxygen species (ROS) may form in the body under the influence of exogenous factors, such as high temperature, ionizing radiation, tobacco smoke, alcohol, and pesticides, and endogenously mainly in the mitochondrial respiratory chain, as well as in reactions involving oxidoreductases, phagocyte activation, purine nucleotide metabolism, and microsomal hydroxylation cycle (cytochrome P-450) [8]. Damage to the genetic material by ROS is the first stage of carcinogenesis; therefore, chronic inflammatory processes associated with the increased production of ROS may be the basis for the formation of cancer [9]. It has been shown that at an early stage of PCa development cancer cells are subject to strong oxidative stress, the level of which is associated with the appearance and progression of the neoplastic process [10,11]. The indirect effect of ionizing radiation on cells, responsible for 75% of their damage, is the generation of ROS, mainly through water radiolysis. The level of intracellular ROS increases after exposure to ionizing radiation and persists for many hours [12]. The cell’s antioxidant apparatus, present in both normal and cancer cells, can reduce ROS levels and repair damage to biomolecules caused by ionizing radiation. This effect is beneficial in the case of normal cells, however, the presence of an efficient antioxidant system may be a factor that reduces the radiosensitivity of the tumor.

Living organisms have intracellular mechanisms that maintain the reduction and oxidation potential (i.e., redox buffer)—a set of enzymatic antioxidants that also includes repair enzymes, e.g., by reducing the products of lipid peroxidation such as type 1 paraoxonase (PON1). The activity of PON1 in plasma shows significant differences between individuals, even several times [13,14,15,16]. More than 200 single nucleotide polymorphisms are responsible for approximately 60% of the individual differences in the activity of this enzyme [17]. PON1 activity is also affected by environmental factors, such as diet, medication, smoking, oxidative stress, and inflammation [13,14,15,17].

The initial study of PONs was related to the toxicological aspects of this enzyme due to the fact of their ability to detoxify organophosphate compounds. More recent studies have demonstrated a correlation of the PON activity with various cardiovascular diseases, such as diabetes, neurological disorders, and cancers [18,19]. This enzyme exerts pleiotropic effects, including a protective effect against the lipid peroxidation of plasma lipoproteins, as well as cell membranes, and protection against the protein homocysteinylation of lipoproteins. A higher PON1 activity is associated with a more effective ability to remove harmful compounds that formed as a result of oxidative stress and, thus, with a lower risk of cardiovascular diseases and cancer [20,21,22].

The severity of oxidative stress depends on the rate of the formation of free radicals and the rate of their removal. PON1 is an element of the so-called third line of defense—mechanisms that remove the products of free radical reactions by repairing or eliminating damaged particles.

In recent years, the serum PON1 activity has been reported to be reduced in several types of malignancies, such as lung cancer, gastrointestinal cancer, breast and prostate cancer, bladder cancer, central nervous system tumors, non-Hodgkin’s lymphoma, and acute lymphoblastic lymphoma [23,24]. Mitochondrial DNA, due to the lack of histones, a less efficient repair system and a continuous method of transmitting genetic information (lack of introns), is more sensitive to the damaging effects of ionizing radiation than nuclear DNA [25]. Damage to mitochondria, which are the most important energy-providing organelles, may result in the increased generation of free radicals, leading to cell damage or death [26,27]. The overexpression of enzymes in the PON family prevents mitochondrial dysfunction; they are linked to mitochondrial membranes, modulate mitochondrial metabolism, and prevent apoptosis. It is also suspected that cancer cells can scavenge serum PON1 and take advantage of its antioxidative effects [28]. Therefore, it is suggested that PON1 may regulate the severity of oxidative stress by protecting mitochondria, thus affecting the radiation sensitivity [29].

The aim of this study was to investigate the changes in the PON1 activity under the influence of ionizing radiation, as well as the relationship between the serum PON1 activity and the degree of advancement of PCa in patients undergoing radiotherapy. We also conducted an analysis of the possible relationships between the PON1 activity and known risk factors for PCa recurrence—PSA and Gleason score, as well as prostate volume, body mass index (BMI), and adipose tissue thickness.

## 2. Materials and Methods

### 2.1. Study Participants

Eighty-four men (mean age: 67.6 ± 13.2 years, range: 53–80) undergoing RT for prostate cancer in 2005–2010 at the Department of Brachytherapy in our hospital were included in the study. All patients had biopsy-proven prostate cancer (clinical stage: T1-T3bN0M0; no metastases in regional lymph nodes; and no distant metastases) and PSA determination. The stage (TNM) was determined according to the American Joint Committee on Cancer (AJCC) TNM classification from 2002 [30]. All patients had pelvic MR (magnetic resonance) before treatment, and the prostate volume was measured using transrectal ultrasonography (TRUS). On the basis of the tests performed, the patients were assessed as belonging to low-, medium-, or high-risk groups, according to d’Amico et al. [31]. These risk groups were based on known prognostic factors: PSA level, biopsy Gleason score, and 2002 AJCC T stage. The patients with an AJCC clinical T stage of T1c and T2a, a PSA level of 10 ng/mL or less, and a biopsy Gleason score of 6 or less were identified to be at low risk (<25% at 5 years) for post-therapy PSA failure. The patients with an AJCC stage T2c disease, a PSA level of more than 20 ng/mL, or a biopsy Gleason score of 8 or more had a risk higher than 50% at 5 years of post-therapy PSA failure. The remaining patients with PSA levels higher than 10 and 20 ng/mL or lower, a biopsy Gleason score of 7, or AJCC clinical stage T2b were found to have an intermediate risk (25–50% at 5 years of post-therapy PSA failure).

The control group consisted of 60 healthy men (without diagnosed PCa or any other malignancy) in whom the PSA concentration and PON1 activity were determined. The mean age in this group was 63.0 ± 13.2 years. Both in the study and in the control group, a similar frequency of comorbidities typical for age was observed: arterial hypertension, type 2 diabetes, and coronary artery disease. 

### 2.2. Brachytherapy 

The patients with clinical stage T1-T2bN0M0, a PSA below 10 ng/mL, and a Gleason score not higher than 6, qualified for HDR brachytherapy alone. Other patients were qualified for combined BT with EBRT. Due to the small number of patients studied, we did not distinguish patients who underwent only BT from those who underwent BT + EBRT in the analysis. Forty-three patients received hormone therapy together with radiation treatment. The radiation schedule was EBRT 46 Gy in 23 fractions (2 Gy/day) combined with two 10 Gy HDR brachytherapy fractions [32].

### 2.3. Determination of the PON1 Activity 

Blood samples were obtained prior to irradiation, after the completion of irradiation, and 7–8 weeks later. The sera were stored at −80 °C until the biochemical analysis, however, no longer than two months. The serum PON1 activity was determined, as we previously described [33], according to Playfer et al. and modified by Sogorb et al., as the rate of hydrolysis of paraoxon at 37° in a TRIS/HCl buffer at pH 10.5, with CaCl_2_, and the activities are expressed as IU/l [34,35].

The study was realized in accordance with the Declaration of Helsinki and approved by the local Ethics Committee (KB/424/2005 with an annex). All participants signed written consent.

### 2.4. Statistical Analysis

The study results were processed using the STATISTICA 8.0 program and are presented as the mean with standard deviations. The statistical analysis was carried out using the Z-test based on a normal distribution to compare the means of two sufficiently large groups (*n* > 50); the parametric test for two structure indicators in order to verify the hypothesis that the frequencies of the distinguished cases in the compared groups did not differ significantly; and Snedecor’s F-test to verify the hypothesis of the homogeneity of the variance in the two compared groups. Parametric tests were applied to compare the mean values. In the case of the homogeneity of the variance in the compared groups, the hypothesis of the equality of the means was verified using the student’s *t*-test. When this condition was not met, the Cochran–Cox test was used. The student’s *t*-test for dependent variables (paired data) or the Wilcoxon test were used when the assumptions of the student’s *t*-test were not met. The nonparametric Shapiro–Wilk test was applied to verify the hypotheses regarding the normality of the distributions of the examined features. If the hypothesis of normal distribution was rejected, the nonparametric Mann–Whitney U test was used to compare the groups. At *p* < 0.05, the difference was considered statistically significant.

Pearson’s linear correlation coefficients, for detecting possible dependencies among the measurable features, and the student’s *t*-test, for verification of the hypothesis regarding the significance of the correlation coefficients, were used.

## 3. Results

### 3.1. Characteristics of the PCa Patients

Ninety-four percent of the patients (*n* = 79) had T1-T2 cancer, and five patients had a T3 disease. A low histopathological stage (Gleason score < 7) was found in 60% of patients. The mean pretreatment PSA was 12.53 ng/mL (2–88 ng/mL), and the mean prostate volume was 34.87 mL (10–64 mL). Fifty-one percent of the patients (*n* = 43) received neoadjuvant hormone therapy. The main clinical characteristics of the patients are presented in Table 1. The average age of the men in the study group was higher (67.6 ± 13.2 vs. 63.0 ± 7.3 years; *p* = 0.013) than in the control group, and the PSA level was significantly higher (12.42 ± 11.78 vs. 1.39 ± 1.12 ng/mL; *p* < 0.001).

### 3.2. PON1 Activity in the PCa Patients

Table 2 shows the PON1 activity in the control and study groups before the start of treatment, immediately after the end of RT (BT HDR + EBRT), and 2 months after the end of treatment. Table 3 show the comparison of the PON1 activity in the study group before the start of treatment, after 2 months (immediately after the end of RT), and after 4 months from the start of treatment (2 months after the end of RT). In the group of PCa patients subjected to BT, a statistically significant decrease in the PON1 activity was observed immediately after RT (by 13%; *p* = 0.006) and 2 months after the end of RT (by 23%; *p* < 0.001) compared to the activity before the start of treatment. The difference between the PON1 activity immediately after BT and after the next 2 months was 5% and was not statistically significant (*p* = 0.542).

Table 4 presents an analysis of the correlation between the PON1 activity and BMI, adipose tissue thickness, prostate volume, PSA concentration, and Gleason score (parameters defined as before the start of RT). There were no statistically significant correlations between the PON1 activity and prostate volume, BMI, adipose tissue thickness, or Gleason score, but a statistically significant relationship was found between the PON1 activity and PSA concentration. The relationship between the PON1 activity and the thickness of adipose tissue at the umbilicus tended to be statistically significant (*p* = 0.064) (Figure 1). There was no statistically significant relationship between the PSA concentration and the PON1 activity in the control group (*n* = 60; r = −0.128; *p* = 0.333).

### 3.3. PON1 Activity in the PCa Patients Receiving and Not Receiving Hormone Therapy

Table 5 shows a comparison of the PON1 activity before RT, 2 months (post-irradiation), and 4 months after treatment in the PCa patients receiving and not receiving hormone therapy. No significant differences in the PON1 activity before or immediately after RT were observed between the patients who underwent and those who did not undergo hormone therapy (*p* = 0.513 and *p* = 0.651, respectively). In the group of patients not receiving hormonal therapy, a statistically significant decrease in the PON1 activity was observed immediately after irradiation (by 14%, *p* = 0.002) and 2 months after the end of RT (by 13%, *p* < 0.018) compared to the activity before the start of treatment (Table 6).

## 4. Discussion

Under the influence of ionizing radiation, the generation of ROS in the irradiated tissue increases, and this effect is largely responsible for the damaging effect of radiation on cells. The damaging effect of ROS and, on the other hand, the activity of the antioxidant system, both inside and outside the cancer cell, are therefore crucial for the effectiveness of RT in destroying cancer cells. However, there are few studies examining changes in the activity of the antioxidant system in patients undergoing BT HDR + EBRT due to the presence of PCa, although there are reports on the relationship between the activity of the antioxidant system and the pathogenesis of this cancer [32,36].

When comparing the study with the control group, attention should be paid to statistically significant differences in the values of two parameters: age and PSA concentration. PCa is a cancer that occurs mainly in older men, with a maximum incidence after the age of 70. The mean age of the study group (i.e., 67.6 years) is a typical age for patients undergoing HDR and EBRT brachytherapy [37,38,39]. The average age of the control group was slightly lower, at 68 years. Such a difference, although statistically significant, should not be a factor that could significantly affect the PON1 activity on its own. Some authors suggest the existence of a relationship between the PON1 activity and age; however, these differences were observed between populations significantly different in age [40]. The PON1 activity may vary in different disease states, the incidence of which is related to the age of the studied population, and this factor may play a much more important role than age alone. In addition, the activity of this enzyme is affected by a number of additional factors, such as genetic mutations, smoking, stress, or a high-fat diet. Some PON1 polymorphisms may contribute to an increased cancer risk associated with pollution and other environmental chemicals.

The first measurement of the PON1 activity in the study group was made before the start of treatment. The comparison of this parameter in the control group with the first result obtained in the study group can be treated as a comparison of the population of healthy men with the population of patients with PCa, without the influence of RT, a factor that may potentially cause a change in the PON1 activity. The results of this comparison did not show statistically significant differences in the PON1 activity, and a small difference (83.96 vs. 84.05 U; *p* = 0.99) suggests that the PON1 activity in patients with PCa and in the population of healthy men was very similar. It can be assumed that PCa does not have the same mechanism of deregulation of PON1 activity as in other cancers.

The obtained results indicate that the activity of PON1 in patients undergoing RT was lower than before treatment and lower than in the population of healthy men. Moreover, this effect persisted 2 months after the end of treatment, and no statistically significant difference was observed when comparing the PON1 activity immediately after the end of RT and after the next 2 months. Serhatlioglu et al., in their work examining the activity of PON1 in workers occupationally exposed to ionizing radiation, obtained similar results; the activity of PON1 in these subjects was significantly lower than in the control group [41]. The authors of this paper did not propose an explanation of the mechanism responsible for the observed effect. This is the only study found that investigates the relationship between the exposure to ionizing radiation and PON1 activity in humans.

Ionizing radiation causes the intensification of oxidation processes in the cell. Antioxidant mechanisms partially eliminate this effect, but they may be weakened in the case of the intensive production of free radicals. It has been shown that ionizing radiation disturbs the oxidant–antioxidant balance by increasing the production of ROS, as well as changing the expression and activity of enzymes responsible for maintaining this balance [36,42,43,44]. Peroxidized lipids formed from lipid compounds under conditions of oxidative stress can inactivate PON1 [45]. These mechanisms may explain the observed decrease in the PON1 activity in patients undergoing RT.

Exposure to ionizing radiation leads to a rapid rise in ROS levels that last for several hours. Damage to the mitochondria, both as a result of the direct action of radiation and the action of free radicals, leads to prolonged oxidative stress. ROS intensify the harmful effect of ionizing radiation, leading to the perpetuation of chronic oxidative stress [36].

Increased exposure to free radicals significantly intensifies the process of lipid peroxidation, which also occurs in the course of normal metabolism. Peroxidized lipids, to a greater extent than ROS with a short half-life, can be mediators of damage to cells adjacent to and even distant from the irradiated site. Thus, even local exposure to ionizing radiation can disrupt the systemic oxidant–antioxidant balance. Not without significance for this balance may also be the processes of fibrosis and reconstruction of the prostate subjected to RT, lasting even for several years after the end of irradiation. In a study by Wozniak et al., the disturbance of the oxidant–antioxidant balance in patients with PCa was observed even up to 2 years after the end of irradiation [32].

Another factor that may influence the differences in the PON1 activity observed in our own research is hormone therapy. A total of 51.2% of the patients in the study group received hormone therapy, and obviously none of the men in the control group received it. Moreover, the first measurement of the PON1 activity in the study group was performed before the possible start of hormone therapy. There were no significant differences in PON1 activity before or immediately after RT between the patients who underwent and those who did not undergo hormone therapy. Only after an additional 2 months, the PON1 activity in patients undergoing hormone therapy was significantly lower than in those not receiving hormonal treatment. However, even after separating the patients not receiving hormone therapy from the study group, there were still significant differences in the PON1 activity before irradiation and immediately after and after the next 2 months. This suggests that although hormone therapy may affect the PON1 activity, it was not the only factor responsible for the observed effect.

PON1 is an element of the antioxidant system, which is responsible for removing free radicals and their effects. On the other hand, the effect of RT on cells depends on the ROS formed as a result of ionizing radiation. The high efficiency of the antioxidant system, which in healthy people provides better protection against the harmful effects of oxidative stress, reducing the risk of cancer, may, in patients undergoing RT, contribute to reducing its effectiveness.

Cancer stem cells, which are considered to be a factor playing an important role in the tumor response to treatment and the recurrence of cancer, show increased resistance to ROS and increased activity of antioxidant enzymes, which means that these cells are more resistant to ionizing radiation [46]. The decrease in the PON1 activity may be an indicator of the number of ROS formed; thus, it could be a measure of the efficacy of RT, as the effectiveness of RT depends on the number of ROS. Therefore, in patients in who the decrease in the PON1 activity after irradiation was greater, a better treatment effect could be expected. Unfortunately, the conducted study does not provide sufficient arguments to prove this thesis; all patients were administered the same dose of radiation; thus, no conclusions can be drawn regarding the possible relationship between the received dose and the reduction in the PON1 activity.

It is puzzling, however, that the size of the groups, which were too small to show the influence of “classic” risk factors for recurrence, was large enough to show the correlation between the PON1 activity and the appearance of recurrence, as we have shown in an already published work [33]. We observed significantly higher PON1 activity in patients who experienced PCa recurrence after RT. This may indicate the important role of the antioxidant system in the response of PCa cells to RT.

No papers examining the relationship between prostate volume and PON1 activity have been found in the literature. The obtained results do not clearly indicate the existence of such a relationship; the significance level of *p* < 0.05 (*p* = 0.096) was not reached. However, this may be due to the small size of the study group. On the other hand, reports have been published indicating the existence of a negative correlation between the BMI and PON1 activity [47]. No such correlation was observed in the study group. This may be related to the sex of individuals in the study group; in a study conducted by Krzystek-Korpacka et al., a statistically significant correlation was observed in girls, while in boys the result was not statistically significant [48].

It is surprising that the level of significance of the correlation between the thickness of the adipose tissue at the level of the umbilicus and PON1 activity approached the limit value of *p* < 0.05 (*p* = 0.061), but the correlation was determined to be positive. The measurement of the thickness of the adipose tissue in MR is a parameter that better reflects the amount of adipose tissue than the commonly used BMI, which is influenced by many factors. If the lack of statistical significance resulted only from too small a sample size and the observed positive correlation exists, it would be in contradiction with the described negative correlation between the BMI and PON1 activity.

PSA is a kallikrein produced by both normal prostate epithelial cells and prostate cancer cells. An increase in the PSA concentration may, therefore, be associated with cancer (a positive correlation with local tumor advancement), as well as an increase in the mass of the gland in the course of BPH or prostatitis. The serum PSA concentration in patients with locally advanced PCa depends mainly on the advancement of the disease and the related change in the histological structure of the prostate, as well as on the biology of the tumor. The less differentiated the tumor cells, the more malignant the tumor; these cells produce less PSA [49]. Therefore, the observed correlation between the PON1 activity and PSA concentration can be explained in several ways. There is a possible relationship between the volume of the prostate (the amount of glandular tissue that produces PSA) and the inflammation of the prostate, which increases the production of ROS, and the activity of PON1. It is worth noting, however, that no significant correlation was observed between the PON1 activity and PSA concentration in the control group, neither was there a correlation between the PON1 activity and prostate volume in the PCa patients. This argues against such an explanation, rather suggesting that the observed correlation is related to cancer.

There was no significant relationship between the PON1 activity and the Gleason score, which is related to the histopathological differentiation of the tumor. The obtained results do not allow for a full explain of the cause of the observed correlation between the PON1 activity and PSA concentration in the study group.

One of the limitations of the study was the small size of the patient subgroups. For this reason, not all patients underwent a second (2 months after the start of irradiation) and a third (after another 2 months) determination of the PON1 activity. Moreover, the correlations between the PON1 activity and BMI, adipose tissue thickness, prostate volume, PSA concentration, and Gleason score were assessed, only taking into account the parameters determined before the start of RT. After the treatment was completed, a series of follow-up visits to the Radiotherapy Outpatient Clinic was planned, during which only a physical examination and subjective examination were performed, as well as the PSA concentration and PON 1 activity in the serum.

In the patients included in the study group, the parameters of lipid metabolism that could potentially affect the PON1 activity were not determined, as these tests are not routinely performed in the observation of patients undergoing brachytherapy for PCa [50]. PON1 is an enzyme associated with the HDL fraction, the activity of which is dependent on oxidized LDL [51]. The PON1 activity may, therefore, depend on the parameters of lipid metabolism and the level of lipid fractions.

Despite these limitations, we managed to obtain statistically significant differences between the subgroups in terms of the PON1 activity.

It would be beneficial if the scope of the performed tests was extended to include further determinations of the PON1 activity after more than 4 months from the start of irradiation, which would allow for the assessment of changes in the PON1 activity in a more distant time after RT and to determine whether and after what time after RT the activity of PON1 normalizes. At the time of designing the experiment, there were no reports on changes in the PON1 activity under the influence of ionizing radiation; it was difficult to predict how long a possible change in the activity of this enzyme might persist after the end of treatment.

## 5. Conclusions

No differences in the PON1 activity were observed between the patients with PCa and healthy men. Exposure to ionizing radiation during brachytherapy and teleradiotherapy caused a decrease in the PON1 activity. There were no significant correlations between the PON1 activity and Gleason score, prostate volume, BMI, or adipose tissue thickness. However, there was a positive correlation between the PON1 activity and PSA concentration in the group of PCa patients. There is a need to expand preclinical radiobiological studies to understand the role and importance of PON1 in radiotherapy treatment.

## Figures and Tables

**Figure 1 antioxidants-12-00212-f001:**
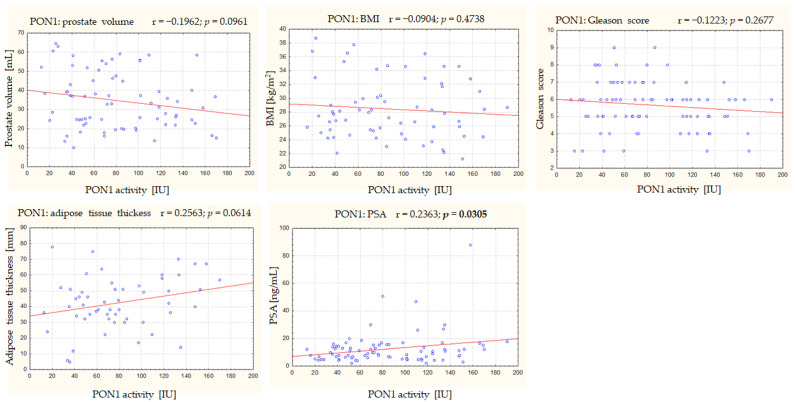
Scatter plots showing the relationships between the PON1 activity and BMI, adipose tissue thickness, prostate volume, PSA concentration, and Gleason score. BMI—body mass index (kg/m^2^); PON1—paraoxonase type 1; PSA—prostate-specific antigen; r—Pearson’s correlation coefficient.

**Table 1 antioxidants-12-00212-t001:** Characteristics of the PCa patients.

	Study Group
*n*	84
Age (mean ± SD) (years)	67.6 ± 13.2
BMI (mean ± SD) (kg/m^2^)	28.2 ± 4.4
Adipose tissue thickness (mean ± SD) (mm)	45.21 ± 16.04
TNM	
T1	20.2% (*n* = 17)
T2	73.8% (*n* = 62)
T3	6.0% (*n* = 5)
Gleason score	
<7	75.0% (*n* = 63)
7	15.5% (*n* = 13)
>7	9.5% (*n* = 8)
PSA (mean ± SD, range) (ng/mL)	12.42 ± 11.78 (2–88)
PSA	
<10 ng/mL	51.2% (*n* = 43)
10–20 ng/mL	40.5% (*n* = 34)
>20 ng/mL	8.3% (*n* = 7)
Risk group	
Low	39.3% (*n* = 33)
Intermediate	42.8% (*n* = 36)
High	17.9% (*n* = 15)
Prostate volume (mean, range) (mL)	34.7 (10–64)
Therapy applied	
HDR BT	42.8% (*n* = 36)
HDR BT + EBRT	6.0% (*n* = 5)
Hormone therapy + RT	51.2% (*n* = 43)

PSA—prostate-specific antigen; SD–standard deviation.

**Table 2 antioxidants-12-00212-t002:** Serum PON1 activity in the control and PCa patients before and after radiotherapy.

	Control Group	PCa Patients		
		Before BT	Immediately after BT	2 Months after BT
*n*	60	84	70	55
PON1 (IU)	83.96 ± 34.52	84.05 ± 43.53	75.72 ± 51.63	59.75 ± 37.95
Z-test		*p* = 0.997	*p* = 0.281	*p* < 0.001

The results are presented as the means ± SD. PON1—paraoxonase type 1; BT—brachytherapy.

**Table 3 antioxidants-12-00212-t003:** Serum PON1 activity in the PCa patients before RT, immediately after the end of irradiation, and 2 months after BT (student’s *t*-test).

	PCa Patients		
	Before BT	Immediately after BT	*p*
*n*	70	70	
PON1 (IU)	86.32 ± 42.38	75.72 ± 51.63	0.006
	Before BT	Two months after BT	
*n*	55	55	
PON1 (IU)	77.54 ± 39.74	59.75 ± 37.95	<0.001
	Immediately after BT	Two months after BT	
*n*	53	53	
PON1 (IU)	57.71 ± 36.12	60.27 ± 37.97	0.542

The results are presented as the means ± SD. PON1—paraoxonase type 1; BT—brachytherapy.

**Table 4 antioxidants-12-00212-t004:** Correlations between the PON1 activity and BMI, adipose tissue thickness, prostate volume, PSA, and Gleason score in the PCa patients.

	PON1 vs. BMI	PON1 vs. Adipose Tissue Thickness	PON1 vs. Prostate Volume	PON1 vs. PSA	PON1 vs. Gleason Score
*n*	65	54	73	84	84
r	−0.090	0.256	−0.196	0.236	−0.122
Student’s *t*-test	*p* = 0.474	*p* = 0.061	*p* = 0.096	*p* = 0.031	*p* = 0.268

BMI—body mass index (kg/m^2^); PON1—paraoxonase type 1; PSA—prostate-specific antigen; r—Pearson’s correlation coefficient. The parameters were defined as before the start of RT.

**Table 5 antioxidants-12-00212-t005:** Serum PON1 activity in prostate cancer patients receiving and not receiving hormone therapy: before radiotherapy, immediately after the end of irradiation and 2 months after BT (Mann-Whitney test).

	PCa Patients		
	Hormone Therapy	Without Hormone Therapy	*p*
*n*	33	39	
PON1 [IU] Before BT	90.7 ± 46.3	83.6 ± 38.4	0.51
PON1 [IU] Immediately after BT	81.1 ± 56.9	71.2 ± 47.0	0.65
PON1 [IU] 2 month after BT	45.9 ± 31.6	69.7 ± 39.4	**0.01**

Results are presented as means ± SD. PON1–paraoxonase type 1, BT–brachytherapy.

**Table 6 antioxidants-12-00212-t006:** Serum PON1 activity in the PCa patients not receiving hormonal therapy before RT, immediately after the end of irradiation, and 2 months after BT (Wilcoxon test).

	PCa Patients Not Receiving Hormonal Therapy		
	Before BT	Immediately after BT	*p*
n	38	38	
PON1 (IU) Mean ± SD Median	82.49 ± 38.29 71.42	71.19 ± 47.04 62.58	**0.002**
	Before BT	Two months after BT	
*n*	32	32	
PON1 (IU) Mean ± SD Median	79.72 ± 37.99 68.77	69.73 ± 39.40 64.58	**0.018**

The results are presented as the means ± SD. PON1—paraoxonase type 1; BT—brachytherapy.

## Data Availability

The data generated or analyzed during this study are included in this article. Data supporting the results of this study, for ethical reasons, are available upon request from the corresponding author.

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
