# Peer review of "Serum Paraoxonase-1 Activity in Prostate Cancer Patients Treated with Brachytherapy as a Measure of Irradiation Efficacy"

_antioxidants, 2023, doi:10.3390/antiox12020212_

Round 1

Reviewer 1 Report

General comments

The authors of this work investigated the effect of radiotherapy on serum antioxidant paraoxonase-1 (PON1) levels in patients with prostate cancer. Correlation between PON1 levels before and after radiotherapy were evaluated finding differences and correlation with other parameters.

The study is interesting since, monitoring of serum levels of PON1, may represent a prognostic and predictive biomarker for patients with prostate cancer. Moreover, these discoveries could open promising perspectives on the study of radiobiological effects in relation to cellular detoxification systems from ROS.

Specific comments

1.       The introduction should be supplemented with a more accurate explanation of the PON1 enzyme. It is the core of the work, therefore, its functions and biological mechanisms involved in ROS detoxification need to be explained in more detail. I suggest adding a few words about the radiotherapy modality of brachytherapy in addition to the information given in the materials and methods.

 2.       The authors showed data reporting a correlation between the activity of PON1, PSA and PON1 and adipose tissue. 1) What are the time points used for the analysis of these correlations? 2) Were they performed immediately, 2 months or 4 months post treatment? 3) Were the same analyzes also performed in relation to hormone therapy? Clarify and argue these points also in discussions.

 3.       My major concern about the work is the title. It makes much emphasis on the relationship between serum PON1 levels and the measure of irradiation “quality”. However, while demonstrating some correlations between PON1 and pre- and post-irradiation PSA levels, the authors present several speculations on the reasons behind this change, in the discussion. At present, also considering the limitations of the study expressed by the authors in the discussion, I would use a less direct title and one that does not so strongly declare a correlation with the quality of irradiation. Rather, I would emphasize the need to enlarge preclinical radiobiology studies aimed at understanding these processes with PON1 to radiotherapy treatment.

 4.       The discussion is full of content that can make scatter the significance of the work. I suggest express the concepts related to the results obtained without digressing excessively with other topics. In fact, some concepts could be reported in the introduction.

 Minor changes

1.       Check the abbreviations and the correct sequence in which they are reported; the full word should be added before the acronym. Check if acronyms can be mentioned in the abstract.

2.       Add some words to explain Gleason scores and TNM clinical stage.

3.       Formatting and alignment of tables should be improved.

4.       The grammatical construction of some sentences could be improved.

Author Response

antioxidants-2141613

Dear Reviewer 1,

Dear reviewer, Thank you for your constructive opinion about my work and for the comments I tried to follow:

1/ The Introduction has been rearranged and supplemented with information on the mechanisms of action of PON1 and the radiotherapy modality of brachytherapy, as suggested.

2/ The correlations between PON1 activity and BMI, adipose tissue thickness, prostate volume, PSA concentration and Gleason score were assessed only taking into account the parameters determined before the start of RT. After the treatment was completed, a series of follow-up visits to the Radiotherapy Outpatient Clinic was planned, during which only physical examination and subjective examination were performed, as well as PSA concentration and PON 1 activity in the serum. This explanation was included among the limitations of the study.

3/ As suggested, we changed the title of the work that does not so strongly declare a correlation with the quality of irradiation. We also emphasized the need to extend preclinical radiobiology studies to understand the role and importance of PON1 in radiotherapy treatment (Conclusions).

4/ Discussion has been rearranged, some of its fragments have also been transposed into the Introduction.

Minor changes

  1. All abbreviations and the correct order in which they are given have been checked, as has the ability to include abbreviations in the summary.
  2. We added an explanation regarding Gleason scores and TNM clinical stage.
  3. Formatting and alignment of tables have been improved.
  4. The manuscript has been checked for accuracy of the English language.

Sincerely,

Dorota Olszewska-SÅ‚onina

Reviewer 2 Report

The paper by Olszewska-Slonina et al. investigated the changes in PON1 activity in 84 patients with prostate cancer undergoing radiotherapy and the relation between serum PON1 activity and degree of advancement of prostate cancer in the same patients. There was no significant difference in PON1 activity between the control group and patients pre-radiotherapy, while there was a decrease in its activity in patients undergoing radiotherapy. They also studied the possible relationship between PON1 activity and known risk factors for prostate cancer recurrence, which are: PSA, Gleason score, prostate volume, BMI and adipose tissue thickness. There were no significant correlations between PON1 activity and these factors, except for PSA concentration which showed a positive correspondence with PON1 in prostate cancer patients.

The article is well written and interesting, hence only some minor concerns should be addressed by the Authors, before a publication on Antioxidants can be granted:

MINOR REVISIONS:

·        The word “quality” in the title is not adequate. Maybe the word “efficacy” is better;

·        Report in the Introduction how PON1 is deregulated in tumors with the references;

·        Always explain an acronym the first time you write it (i.e. MR in line 74);

·        Line 77: please specify the requirements of each risk group for patients’ classification;

·        Line 86: did you distinguish patients who underwent only BT from ones who underwent BT+EBRT in the analysis?

·        Line 104: the words “groups of” before “samples” is missing;

·        Line 132: In the explanation of table 2, you did not write which data are significant and which not;

·        Line 135; rewrite the time-points in the same and in a clearer way throoughout the text and in the captions of the tables and specify which comparisons have been made;

·        Line 149-150: make a separate table and the graph with data of PSA concentration and PON1 activity in the control group (just like table 4);

·        Table 1: add the number of patients who received the different combinations of therapies, the measures of patients’ BMI and adipose tissue thickness;

·        Table 4: PON1 activity considered was the one before or after BT? Specify it in the caption;

·        Add the word “thickness” in the title of third column;

·        Line 173: add the significance of PON1 activity 4 months after BT between patients who underwent and those who did not undergo ormone therapy;

·        Please rephrase the first sentence of the Discussion paragraph;

·        Line 227: consequently, it can be assumed that in prostate cancer there is not the same deregulation of PON1 activity as in other tumors that you cited in line 193;

·        Line 291: replace the word “quality” with “efficacy” and delete the last part of the sentence from “measured” to “depends” because it is a repetition;

·        Line 300: sum up how PON1 activity is related with the appearence of recurrence;

·        Rewrite the sentence in line 319;

·        Line 330-332: where this data can be found?

Overall. MINOR REVISIONS are required.

Author Response

antioxidants-2141613

Dear Reviewer 2,

Dear reviewer, Thank you for your constructive opinion about my work and for the comments I tried to follow:

  • As suggested, we changed the title of the work that does not so strongly declare a correlation with the quality of irradiation.
  • The Introduction has been rearranged and supplemented with information on the mechanisms of action of PON1 and the radiotherapy modality of brachytherapy
  • All abbreviations and the correct order in which they are given have been checked.
  • Line 77: The requirements of each risk group for patients’ classification have been explained, as suggested
  • Line 86: Due to the small number of patients studied, we did not distinguished patients who underwent only BT from ones who underwent BT+EBRT in the analysis. (explained in text)
  • Line 104: The words “groups of” before “samples” is missing - the sentence have been corrected
  • Line 132: In the explanation of table 2, you did not write which data are significant and which not;
  • Line 135: Throughout the text, the description of the time points at which the assays were made was standardized.
  • Line 149-150: Given the limitations on the number of tables and figures in the article imposed by the publisher, data on PSA and PON1 values in the control group are given in the text (PSA - line 130, PON1 activity - Table 2). This also helps to prevent duplication of results.
  • Table 1: The number of patients who received the different combinations of therapies, the measures of patients’ BMI and adipose tissue thickness have been added.
  • Table 4: The correlations between PON1 activity and BMI, adipose tissue thickness, prostate volume, PSA concentration and Gleason score were assessed only taking into account the parameters determined before the start of RT. After the treatment was completed, a series of follow-up visits to the Radiotherapy Outpatient Clinic was planned, during which only physical examination and subjective examination were performed, as well as PSA concentration and PON 1 activity in the serum. This explanation was included among the limitations of the study.

The word “thickness” in the title of third column was added.

  • Line 173: The data on PON1 values 4 months after the start of BT ( it means 2 months after BT) between patients who underwent and those who did not undergo hormone are given in Table 6.
  • Discussion has been rearranged, some of its fragments have also been transposed into the Introduction.
  • Line 227: As suggested, the assumption has been added.
  • Line 291: The sentence has been revised.
  • Line 300: We briefly explained and referenced to our published work focusing on PON1 activity and PCa relapse risk, suggesting that the determination of PON1 activity might be a valuable tool for the prediction of PCa recurrence after RT.
  • Line 330-332: “There was no statistically significant relationship between PSA concentration and PON1 activity in the control group (n=60, r=-0.128, p=0.333)” on line 149-150. The sentence was supplemented with an additional explanation.

Sincerely,

Dorota Olszewska-SÅ‚onina